# Exploring the Impact of Infusion Parameters and In Vitro Digestion on the Phenolic Profile and Antioxidant Capacity of Guayusa (*Ilex guayusa* Loes.) Tea Using Liquid Chromatography, Diode Array Detection, and Electrospray Ionization Tandem Mass Spectrometry

**DOI:** 10.3390/foods13050694

**Published:** 2024-02-24

**Authors:** Hasim Kelebek, Hatice Kubra Sasmaz, Ozge Aksay, Serkan Selli, Ozan Kahraman, Christine Fields

**Affiliations:** 1Department of Food Engineering, Faculty of Engineering, Adana Alparslan Turkes Science and Technology University, 01250 Adana, Turkey; hkelebek@atu.edu.tr (H.K.); haticemedine95@hotmail.com (H.K.S.); ozgegurler2@gmail.com (O.A.); 2Department of Food Engineering, Faculty of Engineering, University of Cukurova, 01330 Adana, Turkey; sselli@cu.edu.tr; 3Applied Food Sciences Inc., 675-B Town Creek Road, Kerrville, TX 78028, USA; cfields@appliedfoods.com

**Keywords:** extractions, antioxidant capacity, LC–MS/MS, Guayusa tea, in vitro digestion, antimicrobial activity

## Abstract

Guayusa tea is derived from the leaves of the *Ilex guayusa* Loes. plant, which is native to the Amazon rainforest. Beyond its pleasant sensory properties, Guayusa tea is rich in antioxidants, phenolics, and minerals. In this study, the effects of infusion time, temperature, and solvent conditions on the color, antioxidant capacity, total phenolic content, phenolic profile, and antimicrobial activity of Guayusa (*Ilex guayusa* Loes.) tea were investigated. Guayusa tea samples were prepared using two different solvents, ethanol and water, with 4, 6, and 8-h infusions at 60 and 70 °C. Liquid chromatography, diode array detection, and electrospray ionization tandem mass spectrometry (LC-DAD-ESI-MS/MS) were used to determine a comprehensive profile of phenolic compounds and to detect differences due to infusion conditions. Moreover, after the Guayusa tea infusion with the highest bioactive properties was determined, the effects of in vitro gastrointestinal digestion on the total phenolic content, antioxidant capacity, and phenolic compounds of the Guayusa tea infusion were measured. Phenolic profile analysis identified 29 compounds, among which chlorogenic acid and its derivatives were predominant. The increase in infusion time was correlated with an elevation in total phenolic content. Significant differences were observed between water and ethanol infusions of Guayusa in terms of phenolics and antioxidants. The total amount of phenolic compounds in the samples prepared with both solvents was found to increase after oral intake, depending on the digestion stage; meanwhile, the amounts of flavonoid compounds and di-O-caffeoylquinic acid derivatives decreased during digestion.

## 1. Introduction

Tea, a beverage with a rich cultural history, has garnered attention for its diverse flavors and potential health benefits attributed to bioactive compounds. Guayusa (*Ilex guayusa* Loes.), a traditional Amazonian tea, has recently gained interest for its unique flavor profile and purported health-promoting properties. Guayusa is an evergreen South American tree that grows between southern Colombia and northern Peru, especially in the upper Amazon basin of Ecuador. It has been used since ancient times for medical benefits such as pain relief and preventing adverse central nervous system effects, especially by Amazonian indigenous tribes [1,2,3]. Commercially, this interest is expected to drive the global herbal tea market to reach a Compound Annual Growth Rate (CAGR) of 4.5%, as estimated for 2018–2024 [4]. Noteworthy for its secure use in traditional foods, Guayusa tea has been designated Generally Recognized as Safe (GRAS) by the United States Food and Drug Administration (GRN 869, 870, 883) [5].

Scientific findings have shown that Guayusa leaf extracts have antioxidant, anti-glycemic, antifungal, antibacterial, and anti-inflammatory effects [2,6]. Studies have reported that caffeine, theobromine, and theophylline concentrations are present in leaf alcoholic extracts, and caffeine and chlorogenic acids were found to be the major compounds [7,8]. The phenolic and carotenoid contents of Guayusa teas were analyzed, and 14 phenolic and 5 carotenoid compounds were identified. It was found that chlorogenic acid and its derivatives from phenolic acids and quercetin-3-hexose, as well as derivatives from flavonoids, were dominant. The researchers also reported that Guayusa leaves were rich in α-carotene, luteolin, and violaxanthin + neoxanthin [2].

A critical factor influencing the bioactive composition of tea is the infusion process, where various parameters such as temperature, time, and solvent can significantly impact the extraction of phytochemicals. The properties of most phenolic substances vary from polar to non-polar in nature; many extraction factors, especially the solvent system, are very challenging for extraction efficiency [3,9]. Therefore, understanding the phenolic profile of Guayusa tea and its alterations under different infusion conditions (time, temperature, type of solvent, etc.) is important for optimizing its health-promoting attributes. Phenolic compounds are also affected by the digestion process and can be released under the action of enzymes, and studies have reported that some flavonoid compounds are released during in vitro digestion. Phenolic compounds have been found to be mainly released in the digestive stages of the stomach and small intestine. Digestion processes are an important factor affecting the biological activity of phenolic compounds in the body [10]. Moreover, the often-neglected aspect of how these compounds transform the digestive process holds pivotal significance, as it dictates their bioavailability and consequential physiological impacts. However, no comprehensive study has investigated the effects of infusion conditions on the phenolic compositions of Guayusa teas.

This study aims to investigate both water and alcohol-based extractions to provide a comprehensive characterization of Guayusa tea leaves. While water-based extracts are common, we aimed to identify phenolic compounds that are insoluble in water but soluble in ethyl alcohol. This study aims to delve into the effects of the intricate interplay between infusion parameters and in vitro digestion on the phenolic profile and antioxidant capacity (2,2-Diphenyl-1-picrylhydrazyl (DPPH) assay, and 2,2′-azino-bis (3-ethylbenzothiazoline-6-sulfonic acid) (ABTS) assay) of Guayusa tea. In the scope of this study, the Guayusa tea infusion with the highest bioactive properties was identified, and the effects of infusion conditions on total phenolic content, antioxidant capacity, and phenolic compounds during in vitro gastrointestinal digestion were evaluated.

## 2. Materials and Methods

### 2.1. Standards and Chemicals

The HPLC standards for chlorogenic acid, caffeine, neochlorogenic acid, quercetin, kaempferol, 3-o-feruloylquinic acid, and kaempferol 3-glucoside were purchased from Sigma Co (St. Louis, MO, USA). Acetonitrile, formic acid, 2,2′-azino-bis-(3-ethyl-benzothiazoline-6-sulfonic acid) diammonium salt (ABTS), Trolox ((±)-6-Hydroxy-2,5,7,8-tetramethylchromane-2-carboxylic acid), gallic acid, sodium carbonate, potassium persulfate, and 2,2-diphenyl-1-picryl hydrazyl (DPPH) were obtained from Merck (Gernsheim, Germany). α-amylase (EC 3.2.1.1), pepsin (EC 3.4.23.1), pancreatin (EC 232.468.9) and bile salts, potassium chloride, sodium bicarbonate, potassium dihydrogen phosphate, sodium chloride, magnesium chloride, ammonium carbonate, and calcium chloride were purchased from Sigma Co. (St. Louis, MO, USA). The standards were prepared daily, and all chemicals used were analytical grade or higher.

### 2.2. Sample Preparations

Guayusa tea infusions were prepared using a water bath (Labotec, Johannesburg, South Africa) with different solvents at varying temperatures and durations. This study involved weighing 3 g of the sample and adding 30 mL of water, which was then infused at 60 °C for 4, 6, and 8 h. The same infusion time intervals were also repeated at 70 °C. Additionally, samples were infused with ethanol (70%) using the same time intervals and temperatures. After each infusion period, the tea was rapidly cooled to room temperature, centrifuged at 6500 rpm for 15 min at 4 °C, and filtered through filter paper (Whatman #1). Each infusion was then further filtered through a 0.20 μm PTFE syringe filter (26 mm, 0.20 μm, Phenomenex, Bologna, Italy) and used for analyses [11,12].

### 2.3. Color Analysis

The color of tea infusions was measured using a Konica Minolta CM-5 (Konica Minolta Optics Inc. Osaka, Japan) colorimeter. The results were presented using five color parameters (L*, a*, b*, C, and h). The L* indicates brightness (whiteness/lightness/darkness), a* represents redness (positive values) or greenness (negative values), b* indicates yellowness (positive values) or blueness (negative values), C denotes the chroma value, and h° signifies the hue angle value [13].

### 2.4. Analysis of Phenolic Compounds by LC-DAD-ESI-MS/MS

Phenolic compound analysis was conducted following the procedure outlined by Kelebek et al. [14] utilizing LC-DAD-ESI-MS/MS in the negative ionization mode. Tea infusion samples were passed through a 0.45 µm membrane filter and directly injected into a HPLC system (Agilent 1100, Agilent Technologies, Palo Alto, CA, USA) connected to a Windows NT 4.0-based ChemStation software system. The LC–MS system employed in this study included a binary pump, degasser, and autosampler. A Phenomenex reversed-phase C-18 column (4.6 mm × 250 mm, 5 μm) (Torrance, CA, USA) was employed, along with a diode array detector (DAD). Two distinct solvents were used: water/formic acid (99:1; *v*/*v*) as Solvent A, and acetonitrile/solvent A (60:40; *v*/*v*) as Solvent B. Phenolic compounds were eluted under the following conditions: 0.5 mL/min flow rate at 25 °C; isocratic conditions from 0 to 5 min with 0% B; gradient conditions for all the following steps: from 0% to 5% B in 20 min, from 5% to 15% B in 18 min, from 15% to 25% B in 14 min, from 25 to 50% B in 31 min, and from 50 to 100% B in 3 min; and washing and reconditioning of the column. Curves were generated using commercial standards, with extracts ranging from approximately 1 to 200 mg/L concentrations and R^2^ values exceeding 0.99. When a reference compound was unavailable, analogous elements were calibrated alongside the molecular weight correction factor. Limits of quantification (LOQ) and detection (LOD) were calculated based on signal-to-noise ratios of 10 and 3, respectively [14].

### 2.5. Antioxidant Capacity and Total Phenolic Content Analysis

The antioxidant capacity was assessed using two distinct methods: 2,2-diphenyl-1-picrylhydrazyl (DPPH) and 2,2′-azino-bis (3-ethylbenzothiazoline-6-sulphonic acid (ABTS) assays. The samples’ DPPH and ABTS-scavenging activities were evaluated according to the procedures outlined in the work completed by Kelebek et al. [15]. In both antioxidant capacity analyses (DPPH and ABTS), a Trolox standard solution was used at various concentrations to obtain the standard curve (3.125–200 µmol). The results were expressed as mM TE (Trolox equivalent)/L.

The total phenolic content (TPC) of tea infusion samples was analyzed via the Folin–Ciocalteu method [16]. A total of 200 µL of extract/standard solution and 1.5 mL of Folin–Ciocalteu reagent (1:10 ratio) were added to the spectrophotometer cuvette. Five minutes later, 1.5 mL of 6% sodium carbonate solution was added, and the mixture was kept in the dark at room temperature for 90 min. The absorbance values were then measured at 765 nm using an UV-Vis spectrophotometer (BMG Labtech, Spectrostar Nano, Ortenberg, Germany). Gallic acid was used as a standard. The total phenolic content of the sample was calculated using a calibration curve prepared with the standard, and the results were expressed as mg GAE (gallic acid equivalent)/L.

### 2.6. Antimicrobial Activity Test

*Staphylococcus aureus* ATCC 29213, *Bacillus subtilis* ATCC 11774, *Klebsiella pneumoniae* ATCC 13883, and *Escherichia coli* ATCC 25922 were used as test organisms. These microorganisms were spread-plated on tryptic soy agar and incubated at 35 °C for 12–18 h. After incubation, the turbidity of the bacterial suspension was adjusted to 108 CFU/mL (0.5 McFarland unit) with a sterile saline solution. The antimicrobial activities of extracts on the selected test organisms were determined with the agar well diffusion technique [17]. Muller–Hinton agar plates were prepared by spreading 100 µL of the inoculum over the entire agar surface. Then, a 6 to 8 mm diameter well was punched aseptically with a sterile cork borer, and 30 to 50 µL of extract was added to the wells. Agar plates were incubated under suitable conditions (35 °C for 18–24 h), depending on the test microorganism. Each experiment was carried out in duplicate. After incubation, inhibition zones were measured three times, and the average was calculated.

### 2.7. In Vitro Digestion

The in vitro digestion model of Infogest, which includes the sequential simulation of oral, gastric, and intestinal digestion, was performed according to the method described in the study of Brodkorb et al. [18]. The total phenolic content, antioxidant capacity (DPPH and ABTS), and total phenolic substance (LC-DAD-ESI-MS/MS) samples collected from the oral, gastric, and intestinal phases were determined with three replicates.

### 2.8. Statistical Analysis

The results obtained were analyzed with a confidence level of 95% (*p* < 0.05) using the SPSS software program (version 24, SPSS Inc., Chicago, IL, USA), after evaluations for normality and variance homogeneity. Duncan’s multiple comparison test was used to determine whether or not there were significant differences. In addition, a heatmap was used in the XLSTAT programme (trial version, 2023).

## 3. Results

### 3.1. Color Characteristics of Infusions

Color is one of the most powerful aspects influencing the consumer acceptability of herbal teas [19,20]. The impact of varying time, temperature, and solvent conditions on the color of Guayusa tea infusions was determined. The results of the color analysis are provided in Table 1. The color parameters (L*, a*, b*, C, and h°) exhibited significant variations based on infusion times and temperature (*p* < 0.05). It was determined that the L* value gradually decreased with increasing infusion time, and the color became darker. The L* value was low in infusions prepared with water, resulting in a darker color than the ethanolic extracts (Table 1). As the infusion time increased, there was a greater transfer of substances into the water, resulting in a decrease in the L* value, which in turn could have contributed to the darkening of the color. It was also found that phenolic compounds were higher in water-phase extracts with a lower L* value, i.e., a darker color. Therefore, the increasing amount of water-soluble phenolic compounds caused the extract to be darker in color. No study has been conducted in the existing literature to explore the infusion of Guayusa teas employing diverse solvents, altering infusion durations, and varying temperature conditions. Consequently, a comparison with the literature data was not attainable. Previous studies have determined that the L* value decreases as the infusion time increases in black tea infusions, leading to a darker color [13,21]. Another study reported that the L* value decreased with increasing infusion time and temperature in black tea samples, and the color became darker with longer infusion times [12]. Moreover, Uzlasir et al. [11] found that the L* value decreased with infusion time while exploring the phenolic compositions, antioxidant properties, and colour characteristics of elderberry flowers’ methanol, ethanol, and aqueous extracts.

### 3.2. Antioxidant Capacity and Total Phenolic Content Analysis

Herbal infusions have been taken as beverages for generations due to their high antioxidant capacity. These infusions have a lot of promise as important natural sources of antioxidants, with the ability to reduce diseases caused by oxidative stress [22].

Assays utilizing the ABTS radical cation are widely employed in assessing antioxidant capacity, alongside the commonly used DPPH assay. The ABTS method involves the spectrophotometric measurement of changes in the ABTS cation radical concentration resulting from its reaction with antioxidants. Meanwhile, the DPPH assay offers a rapid, straightforward, and cost-effective means of evaluating antioxidant activity. It relies on the reduction of DPPH, a stable free radical compound [23]. The antioxidant capacity of the infusions was evaluated using the DPPH and ABTS methods, and the results are presented in Table 2.

It was found that the antioxidant capacity of Guayusa tea infusion samples increased directly with increasing infusion time and infusion temperature (*p* < 0.05). Among the Guayusa tea infusion samples, the infusion prepared with water exhibited the highest DPPH and ABTS antioxidant capacities. The Guayusa tea infusion sample prepared with water at 70 °C for 8 h (Gw-70 °C-8 h) exhibited the highest antioxidant capacity, with DPPH and ABTS values of 86.12 mM TE/L and 88.19 mM TE/L, respectively. Compared to ethyl alcohol infusions, water infusions showed better antioxidant potential across all time and temperature ranges. In both solvents, increasing the temperature and duration increased the antioxidant potential.

In a study by Pardau et al. [24], antioxidant capacity values for Guayusa, determined by the ORAC method, ranged between 798 and 2019 μmol TE/g. The researchers concluded that Guayusa is a good source of phenolic compounds with antioxidant properties. In another study, Garca-Ruiz et al. [2] assessed the antioxidant capacity of Guayusa green leaves using DPPH and ORAC techniques. The antioxidant capacity of Guayusa green leaves was 32.98 mmol Trolox 100 g/DW according to the DPPH assay and 154.03 mmol Trolox 100 g/DW according to the ORAC test. These findings suggest that Guayusa tea leaves have a high antioxidant capacity. On the other hand, by evaluating the antioxidant capacities of black tea derived from the Camellia sinensis plant under various brewing conditions, the DPPH antioxidant capacity was found to range from 1505.20 to 2454.17 mmol TE/L, while the ABTS analysis yielded values between 1965.14 and 3214.96 mmol TE/L [12]. Comparing our findings from Guayusa tea infusions to those of black tea showed that black tea exhibited a higher antioxidant capacity.

The total phenolic contents (TPCs) of Guayusa tea infusion samples are given in Table 2. Significant differences depended on temperature, infusion time, and solvents (*p* < 0.05). The highest TPC was observed in the Gw-70 °C-8 h (19,467.58 mg GAE/L) sample. The results showed that the TPCs of Guayusa tea infusion samples increased depending on infusion time and temperature. High extraction temperatures increased the permeability of cell walls to solvents and components, thereby increasing extraction efficiency. The solubility of tea components increased with infusion time and temperature [13]. In a study investigating infusion-dependent changes in the phenolic, antioxidant, and color properties of St. John’s wort (*Hypericum perforatum* L.) teas, findings indicated a substantial increase in both phenolic compound levels and antioxidant activity under extended infusion periods across three distinct tea extractions [25]. The elevation in the solubility of tea components could have contributed to the augmentation of both antioxidant capacity and TPC.

Correlation analysis was performed to determine the relationships between antioxidant values (DPPH, ABTS), TPC, and phenolic compounds in samples after different infusion and extraction conditions (Figure 1 and Figure 2). Figure 1 shows a high and positive correlation was found between Gw (r = 0.92 for TPC and DPPH; r = 0.78 for TPC and ABTS) and Get (r = 0.89 for TPC and DPPH; r = 0.93 for TPC and ABTS) infusions, as supported by the research findings (Figure 1 and Figure 2).

Correlation analysis to determine the relationships between antioxidant activity, TPC, and phenolic compounds in Guayusa tea samples is not available in the literature. Rodriguez Vaquero et al. [26] found high correlations between the TPC and DPPH capacities of tea infusions. Furthermore, in a study evaluating the relationship between TPC and the antioxidant capacity of boiled brew and tea plant infusions via the Pearson correlation matrix, TPC was reported to be positively and significantly correlated with DPPH capacity for all infusion times. It was reported that the strongest positive correlation was observed between TPC and DPPH capacity in mint and linden leaf infusions [26,27].

### 3.3. Phenolic Profile of Guayusa Infusions

Data representing the retention time, ʎmax in the UV region, molecular ion, main fragment ions in MS^2^, and tentative compound identification obtained by HPLC-DAD–ESI-MS/MS analyses are presented in Table 3. LC–ESI-MS/MS multiple reaction monitoring (MRM) chromatograms of some of the identified phenolic compounds in Guayusa tea infusions are given in Figure 3, Figure 4, Figure 5 and Figure 6. A total of 29 phenolic compounds were identified and quantified. Similar phenolic profiles were observed in infusions at both temperatures and times, but the amounts of phenolic compounds showed significant increases with increasing time and temperature (*p* < 0.05). An increase in infusion time also led to an increase in total phenolic content. Another important piece of data that draws attention in this study shows that the amount of phenolic compounds is significantly higher in infusions made with water than in infusions made with ethyl alcohol (*p* < 0.05). When phenolic profiles were evaluated, chlorogenic acid and its derivatives (CGAs) were dominant in all infusions. CGAs are esters of hydroxycinnamic acids (HCAs) such as caffeic acid (CFA), ferulic acid (FA), p-coumaric acid (p-CoA), and sinapic acid (SA) to quinic acid (QA or 1L-1(OH),3,4/5-tetrahydroxycyclohexane carboxylic acid). These complex compounds exhibit a wide range of physicochemical properties due to positional esterification on the quinic acid moiety, forming regio-derivatives [28].

Caffeic acid derivatives: Three caffeic acid derivatives were identified at *m*/*z* = 341 (Peaks 1–3). These molecules, identified as caffeic acid hexosides, produced the same fragmentation ions corresponding to hexose moiety loss (162 Da) but with slightly varied abundances. The same [M-H]^−^ at *m*/*z* = 341 is also formed by caffeoyl hexoses, in which caffeic acid is coupled to the sugar moiety by an ester bond rather than an ether bond; however, fragments indicative of sugar moiety fragmentation are seen. The fourth chemical with a *m*/*z* = 341 fragment ion was recognized as a caffeic acid O-glucoside derivative. Caffeic acid glucoside-1 was identified as the predominant compound within this group. Its concentration ranged from 115.30 to 127.78 mg/L in water-based infusions and 117.10 to 140.14 mg/L in ethyl-alcohol-based infusions.

Caffeoylquinic acid derivatives: Guayusa infusions include three positional derivatives of caffeoylquinic acids. Peaks 6–8 in the ESI-MS/MS in negative ion mode produced the same [M-H]^−^ ion at *m*/*z* 353 as the chemical formula C_16_H_18_O_9_ predicted. In MS/MS, the molecular ions [M-H]^−^ produced four peaks at *m*/*z* 191, 179, 173, and 135. Peaks 6, 7, and 8 had the structures of neochlorogenic acid, cryptochlorogenic acid, and chlorogenic acid, respectively. Caffeic acid and quinic acid have been esterified to generate chlorogenic acids. In ESI MS in negative ion mode, the diagnostic fragmentation patterns of chlorogenic acid derivatives included the cleavage of intact caffeoyl and quinic acid fragments. Chlorogenic acid was determined to be the dominant compound in both infusion conditions and varied between 6428.28 and 6557.41 mg/L in water-based infusions and 4484.53 and 5145.29 mg/L in ethyl-alcohol-based infusions. Neochlorogenic acid was the other dominant compound after chlorogenic acid. These two compounds constitute a significant part of the total amount of phenolic compounds. García-Ruiz et al. [2] reported that chlorogenic and neochlorogenic acids were also dominant compounds in Guayusa teas.

One study reported that the chlorogenic acid content of a cup of coffee (200 mL) varied between 20 and 675 mg of chlorogenic acid in the coffee content, but these values varied according to the coffee type and brewing method [29]. Moreover, in another study to determine the chlorogenic acid content of green coffee infusions, it was reported that the amount of chlorogenic acid varied between 628 and 1040 mg/L in C. arabica infusions and between 682 and 1210 mg/L in C. canephora infusions [30]. The data we obtained on the chlorogenic acid amounts of Guayusa tea in water-based and ethanol infusions were higher than in coffee samples, compared to previous studies.

Coumaroylquinic acid derivatives: The hierarchical scheme keys for the LC-MS^n^ identification of CGAs were used to identify the metabolite with a molecular ion [M-H]^−^ at *m*/*z* 337, identified as 5-coumaroylquinic acid, because it formed a fragment ion at *m*/*z* 191 [QA-H]- indicating loss of a coumaroyl moiety (Peak 4, Peaks 8–12) (Table 3). Six compounds were identified in the structure of the coumaroylquinic acid isomer, with p-Coumaroylquinic acid-2 identified as the dominant one. These compounds increased with rising time and temperature, reaching higher concentrations in ethyl-alcohol-based infusions than in water infusions (Table 4 and Table 5).

Feruloylquinic acid derivatives: The same approach was used to identify two feruloylquinic acid (FQA) derivatives (Peaks 13 and 14), which were recognized by their precursor ion [M-H]^−^ at *m*/*z* 367 and based on the fragmentation patterns and Rt given in Table 3. Despite their differing strengths, the two FQA regio-derivatives were discovered in Guayusa infusions. The base peaks at *m*/*z* 193 [FA-H]- and *m*/*z* 173 [QA-H-H2O]- were employed as diagnostic peaks for 3-FQA and 5-FQA, respectively, as specified in the hierarchical scheme keys for the LC-MS^n^ identification of CGAs. FQA yields *m*/*z* 134 [FA-H-CO_2_-CH_3_]^−^ as well. As a result, Molecules (13) and (14) were labeled as 3-FQA acid and 5-FQA, respectively. The amounts of the 3-FQA compound ranged between 70.69 and 119.38 mg/L, and that of the 5-FQA compound ranged between 72.51 and 130.08 mg/L in water-based infusions. In ethyl-alcohol-based infusions of these compounds, the amounts were found between 85.88 and 129.19 mg/L and between 103.91 and 165.17 mg/L, respectively. Ethyl alcohol infusions provided higher solubility.

Dicaffeoylquinic acids: Three di-CQA compounds were detected in the prepared tea infusions. These molecules exhibit spectrum features with UV maxima at 242.6 and 327.0 nm and retention times (Rt) of 50.53, 52.37, and 54.85 min. The ESI-MS/MS spectra revealed fragment ions [M-H]^−^ at *m*/*z* 515, [M-C_9_H_6_O_3_] at *m*/*z* 353, and [M-H-2C_9_H_6_O_3_] at *m*/*z* 191 (Table 3). The identification of these compounds, which exhibited identical spectral data, was confirmed using standard substances. 3,4-di-O-caffeoylquinic acid, 3,5-di-O-caffeoylquinic acid, and 4,5-di-O-caffeoylquinic acid were found to be the dominant compounds in the study, and it was determined that the infusions prepared with ethyl alcohol were richer in terms of these compounds.

Flavonoids: Twelve flavonols were detected in tea infusions, with quercetin derivatives found as the predominant flavonols in all samples. Seven peaks were identified as quercetin derivatives based on their UV spectra and MS fragmentation, ultimately producing the quercetin aglycone at *m*/*z* 301 in negative mode (Table 3). There are some peaks with the same [M-H]^−^, as shown in Table 3. Peaks 19 and 20 both exhibited [M-H]^−^ at *m*/*z* 463, resulting in a fragment at *m*/*z* 301 (hexose moiety loss). The MS^2^ spectra of *m*/*z* 301 yielded quercetin-like ions at *m*/*z* 179 and 151 in both cases. The two peaks were confirmed by comparing their absorption spectra and retention times to those of legitimate standards. Peak 19 was tentatively identified as quercetin 3-O-galactoside, and Peak 20 was tentatively recognized as quercetin 3-O-glucoside. Quercetin 3-O-glucoside was found to be the predominant compound in the group of quercetin-derived compounds, followed by quercetin 3-O-galactoside. The amounts of these two compounds and other quercetin derivative compounds were higher in ethyl-alcohol-based infusions than in water-based infusions (Table 4 and Table 5).

Based on their UV spectra and MS fragmentation, five peaks were recognized as kaempferol derivatives, leading to the kaempferol aglycone at *m*/*z* 285 in the negative mode. The [M-H]^−^ of a kaempferol-hexose conjugate is represented by the ion at *m*/*z* 447, and the resultant MS^2^ fragment at *m*/*z* 285 ([M-H]^−^ 162) is a kaempferol. Peak 25 was identified as kaempferol-3-O-glucoside and had [M-H]^−^ at *m*/*z* 447 with a fragment at *m*/*z* 285 (loss of 162 amu, hexose moiety). Peaks 23 and 25 were also confirmed when their absorption spectra and retention durations were compared to the standards. Peak 25 had the same MS spectra as Peak 23, as indicated in Table 3. Galactosides elute before matching glucosides, and a kaempferol galactoside was discovered in this manner [31].

Five compounds in the structures of kaempferol derivatives were determined. Kaempferol 3-O-galactoside was the dominant compound in water (172.75–287.40 mg/L) and ethyl alcohol (250.18–328.08 mg/L)-based infusions. As with other flavonoid compounds, the amount of this compound was higher in ethyl alcohol infusions, and significant increases were determined depending on increasing time and temperature (*p* < 0.05).

A heatmap was used to visualize the distribution of phenolic compounds in samples after different infusion and extraction conditions. Figure 7 and Figure 8 show heatmaps generated from the data in Table 4 and Table 5, which give an overview of the magnitude of the numeric differences observed in all phenolic compounds in Guayusa samples compared to infusion conditions for each compound. The average concentration of each phenolic was marked by a different color on the heatmap, changing between blue and red. Darker red tones indicate major abundance, while darker blue tones indicate minor quantities. As can be seen in Figure 7 and Figure 8, Guayusa teas were divided into two clusters based on the infusion temperatures of 60 °C and 70 °C, which were grouped into separate categories. Samples clustered in the same category show high similarity and correlation.

As a result of the comprehensive evaluations in this study, which explored different solvents, infusion times, and temperatures, the highest antioxidant capacity and phenolic compound potential were identified in water- and ethyl-alcohol-based infusions at a temperature of 70 °C for 8 h. In the second stage of the study, changes in antioxidant potential and phenolic compounds were investigated during the antimicrobial effect and in vitro digestion stages of these samples.

### 3.4. Antimicrobial Activity

The antimicrobial effects of Guayusa tea infusion against two Gram-positive (*S. aureus* ATCC 29213 and *B. subtilis* ATCC 11774) and two Gram-negative (*K. pneumoniae* ATCC 13883 and *E. coli* ATCC 25922) pathogenic microorganisms were investigated (Table 6). Greater zone diameters indicate a higher antimicrobial effect. As can be seen in Table 6, water-based infusions with an elevated phenolic and antioxidant potential demonstrated a stronger antimicrobial effect compared to ethyl-alcohol-based infusions. It was determined that teas generally exhibited a high antimicrobial effect on various microorganisms, except for *E. coli* ATCC 25922, where no significant effect was observed (*p* > 0.05). Gram-negative bacteria, such as *E. coli,* are more resilient to polyphenols than Gram-positive bacteria due to their distinct cell wall compositions. One study found no antimicrobial activity against *E. coli,* possibly due to a pair of membranes surrounding each bacterial cell and a unique outer membrane that provides some form of resistance to these bacteria [32]. A study investigating the antibacterial activity of green tea, black tea, and different oolong teas reported antibacterial activity against three pathogenic bacteria, namely, *S. aureus* ATCC 29213, *E. faecalis* ATCC 29212, and *P. aeruginosa* ATCC 27853, as well as antifungal activity against *C. albicans* ATCC 10231. Previous studies have shown that green and black tea have antibacterial effects against pathogens, including *E. faecalis*, *S. aureus*, *C. albicans*, and *P. aeruginosa*. Furthermore, green tea has been shown to have antimicrobial effects against both Gram-positive and Gram-negative bacteria (e.g., *E. coli*, *Salmonella* spp., *S. aureus*, and *Enterococcus* spp.) [33,34,35].

### 3.5. Impact of In Vitro Digestion on Bioactive Compound Profiles

The Guayusa tea infusion with the highest bioactive properties was determined (Gw-70 °C-8 h and Get-70 °C-8 h) and the effects of in vitro gastrointestinal digestion on total phenolic content, antioxidant capacity, and phenolic compounds were investigated.

Effect of in vitro digestion on antioxidant capacity and total phenolic content: The antioxidant capacity (DPPH and ABTS) and total phenolic content of the samples at the in vitro gastrointestinal digestion stages are given in Table 7. After simulated in vitro digestion, an increase in DPPH, ABTS, and TPC results was observed. In the infusion sample of Gw-70 °C-8 h, the DPPH antioxidant capacity was 37.82, 45.72, and 158.42 mM Trolox/L, whereas the ABTS antioxidant capacity was 51.58, 61.84, and 66.81 mM Trolox/L in the oral, gastric, and intestinal phases, respectively, as presented in Table 7. Similarly, in the infusion sample of Get-70 °C-8 h, the DPPH antioxidant capacity was 43.34, 46.64, and 61.36 mM Trolox/L, while the ABTS antioxidant capacity was 61.04, 67.08, and 70.01 mM Trolox/L in the oral, gastric, and intestinal phases, respectively, as shown in Table 7.

Bioactive compounds are obtained using different types of solvents that affect the biological properties of the extracts [36]. Ethanol and water are two different solvents used to extract bioactive compounds. These two solvents may have affected in vitro digestion, and their bioavailability may have differed.

Studies have found that the antioxidant capacity varies throughout the process of digestion. This is due to the fact that a significant degree of the radical scavenging capacity of phenolic compounds relies on the pH of the environment. As a result, the antioxidant capacity can fluctuate while undergoing digestion. Additionally, phenolic compounds can experience structural changes while undergoing gastrointestinal transit due to the ionization of hydroxyl groups. This may lead to an increase in antioxidant capacity at higher pH values [37]. Enzymes (e.g., α-amylase, pepsin, and pancreatin), temperature (37 °C), and changes in pH during in vitro digestion (e.g., 2.5 or 7) can affect the release of antioxidant compounds in different ways. These effects depend on the food matrix and its interaction with other compounds such as proteins, carbohydrates, lipids, fiber, or minerals. Enzymes used at various stages of in vitro digestion have been shown to influence the behavior of various molecules and their degradation/formation. These interactions and the degradation/formation of molecules in in vitro digestion play an important role in changing the bioavailability of antioxidants [38,39].

Effect of in vitro digestion on phenolic compounds: The bioaccessibility of phenolic compounds in Guayusa tea infusion was determined by Infogest static in vitro gastrointestinal food digestion simulation. The phenolic compound profiles in oral, gastric, and intestinal samples was determined using HPLC-DAD-ESI-MS/MS and significant differences were observed between oral, gastric, and intestinal samples after simulated in vitro digestion (*p* < 0.05) (Table 8). During the gastrointestinal digestion of Guayusa water and ethanol infusion samples, a total of 27 phenolic compounds were identified, including caffeic acid glucoside, neochlorogenic acid, 3-O-feruloylquinic acid, 5-O-feruloylquinic acid, quercetin rutinoside, kaempferol 3-O-galactoside, 3,4-di-O-caffeoylquinic acid, kaempferol 3-O-glycoside, 3,5-di-O-caffeoylquinic acid, and 4,5-di-O-caffeoylquinic acid (Table 8).

When the oral, gastric, and intestinal digestion phases were compared in both infusion samples, it was found that there was an increase in the amount of total phenolic compounds. However, when a comparison was made according to pre-digestion, it was found that there was an 8.4% decrease in the oral phase, a 6.2% decrease in the gastric phase, and a 2.4% increase in the intestinal phase of ethyl-alcohol infusions. Similar changes were found in infusions prepared with water. There was a 4.7% decrease in the oral phase, a 2.6% decrease in the gastric phase, and a 3% increase in the intestinal phase. The highest bioavailability in water and ethanol infusions was observed in the intestinal phase. After simulated gastric digestion, many phenolic compounds’ (quercetin rutinoside, quercetin derivative, quercetin 3-O-galactoside, quercetin 3-O-glucoside, kaempferol 3-O-rhamnoside, kaempferol 3-O-galactoside, 3,4-di-O-caffeoylquinic acid, kaempferol 3-O-glycoside, 3,5-di-O-caffeoylquinic acid, and 4,5-di-O-caffeoylquinic acid) amounts were found to decrease. Neochlorogenic, chlorogenic, cryptochlorogenic, and p-coumaroylquinic acids increased after the gastric phase. The increase in caffeoylquinic acid derivatives can be attributed to the decrease in di-caffeoylquinic acid compounds resulting from fragmentation and the subsequent formation of these compounds [40]. It has been reported that the stability of di-O-caffeoylquinic acids is low, and depending on pH and temperature conditions, diCQAs can isomerise with each other and transform into mono-CQAs, caffeic acid, and compounds with the formula C_15_H_14_O_6_ [41]. Changes in pH levels and the presence of bile salts during simulated digestion may significantly impact the reduction of the quantity of phenolic compounds and increase the monomeric CQAs [42]. In previous studies, it was observed that most of the polyphenols in herbal infusions decreased markedly during the gastric stage and increased after the intestinal stage. It has been suggested that this may be due to the pH levels and other environmental conditions of the gastric and intestinal phases being unsuitable, resulting in damage to certain polyphenols [43,44]. The study conducted by Ozkan et al. [37] investigated the stability and bioaccessibility of phenolics during digestion by using a static in vitro digestion protocol on plant infusions from different regions of Turkey. The results showed that there was a significant increase in the content of several polyphenols, including gallic acid, protocatechuic acid, epicatechin, chlorogenic acid, rutin, and syringic acid, after in vitro intestinal digestion. However, the study also found that most of the polyphenols in herbal infusions decreased significantly during the gastric phase and increased after the intestinal phase. Some polyphenols were even undetectable after in vitro digestion. 

## 4. Conclusions

This study determined the effects of infusion time, temperature, and solvent type on the color, antioxidant capacity, total phenolic content, and phenolic profile, as well as the impact of in vitro gastrointestinal digestion on the phenolic content, antioxidant capacity, and phenolic compounds of Guayusa tea. In general, when Guayusa tea infusions were evaluated, it was observed that the L* value decreased as the infusion time increased, and the color became darker. Antioxidant capacity and total phenol content analysis showed that Guayusa tea infusions had significant antioxidant properties. DPPH and ABTS results showed that the antioxidant capacity of the infusions increased with increasing infusion time and temperature. The phenolic profile of the infusions was determined and found to be dominated by chlorogenic acid and its derivatives. In Guayusa water infusions, the sample containing the highest phenolic content was found to be Gw-70 °C-8 h, and in ethanol infusions, it was Get-70 °C-8 h. Antimicrobial activity was determined against *Staphylococcus aureus* ATCC 29213, *Bacillus subtilis* ATCC 11774, *Klebsiella pneumoniae* ATCC 13883, and *Escherichia coli* ATCC 25922 test organisms. Antimicrobial activity analysis showed that Guayusa tea infusions generally have high activity against Gram-positive bacteria.

Depending on the digestion stages, the total amount of phenolic compounds in the samples prepared with both solvents increased after oral intake. When the profiles of phenolic compounds were analyzed, neochlorogenic acid, chlorogenic acid, cryptochlorogenic acid, 3-O-feruloylquinic acid, 5-O-feruloylquinic acid, and p-coumaroylquinic acids derivatives increased significantly (*p* < 0.05). The amounts of flavonoid compounds and di-O-caffeoylquinic acid derivatives decreased in the digestion stages.

In conclusion, this study revealed that Guayusa tea has antimicrobial effects, high antioxidant properties, and phenolic compounds. It also aligns with the broader objective of advancing tea science by integrating traditional knowledge with modern analytical techniques. The outcomes are poised to contribute valuable insights to the burgeoning field of functional foods and beverages, with implications for the tea industry and public health. In the future, optimizing infusion parameters and exploring novel extraction techniques could enhance the extraction of bioactive compounds from Guayusa tea. Further characterization of phenolic compounds, elucidation of their mechanisms of action, and assessments of bioavailability are crucial for understanding their health benefits. Additionally, research should focus on sustainable cultivation practices, formulation development, and global market opportunities, while fostering multidisciplinary collaboration and integrating traditional knowledge with modern science to promote cultural appreciation and environmental protection.

## Figures and Tables

**Figure 1 foods-13-00694-f001:**
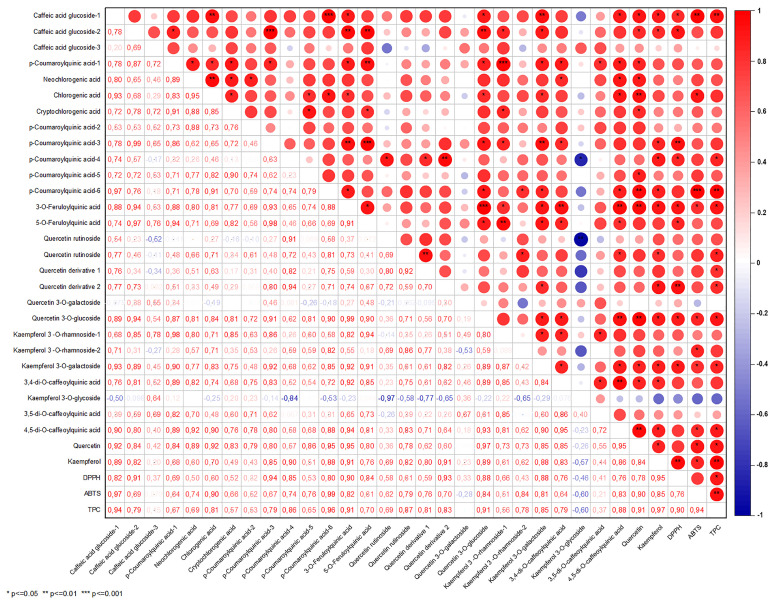
The correlation matrix of the antioxidant activity and phenolic profile of the Guayusa ethanol-water (Get) infusions.

**Figure 2 foods-13-00694-f002:**
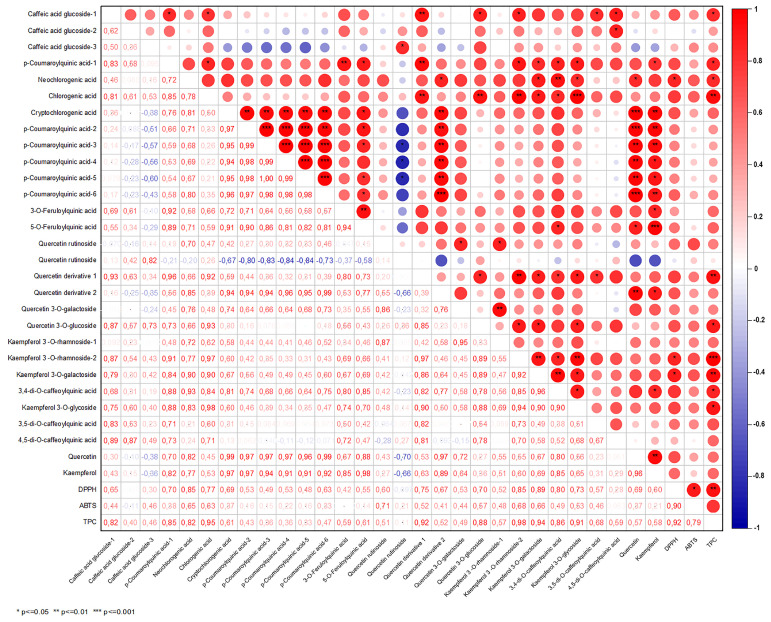
The correlation matrix of the antioxidant activity and phenolic profile of the Guayusa water (Gw) infusions.

**Figure 3 foods-13-00694-f003:**
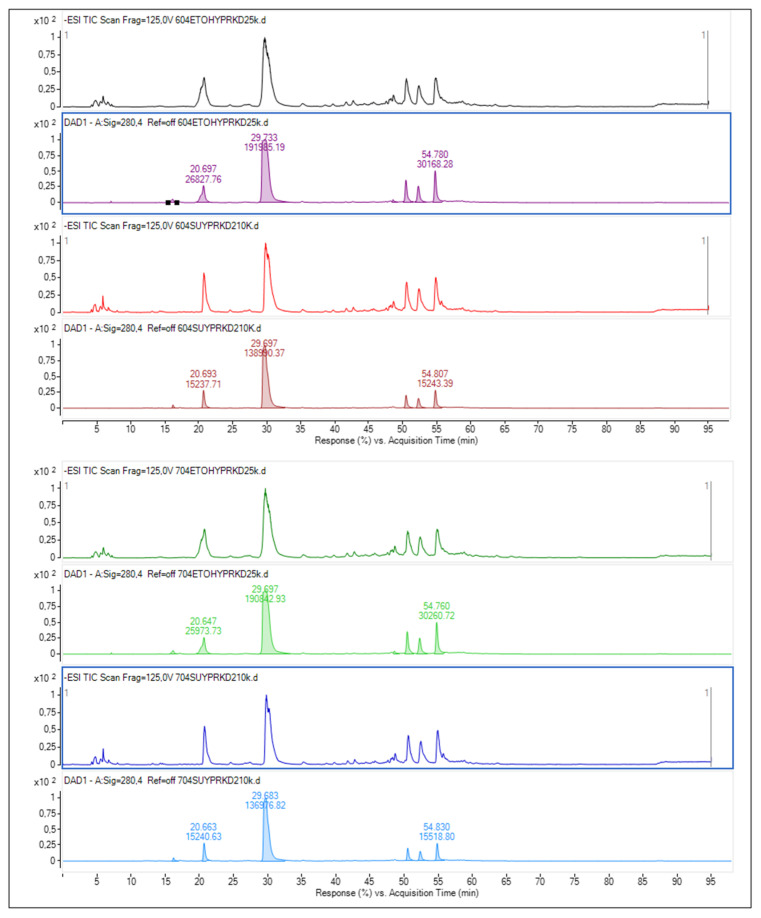
Chromatogram of phenolic compounds identified by LC-DAD and LC–ESI-MS/MS in Get infusions.

**Figure 4 foods-13-00694-f004:**
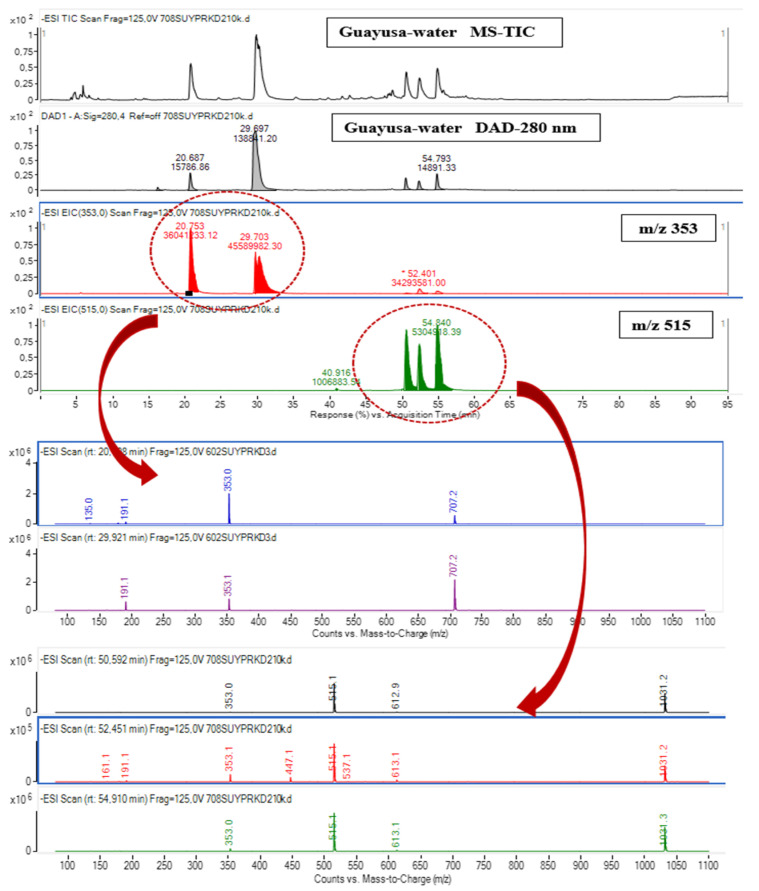
ESI-MS spectrum of chlorogenic acids and di-O-caffeoylquinic acids; LC-MS/MS chromatogram in negative MRM mode with mass transitions.

**Figure 5 foods-13-00694-f005:**
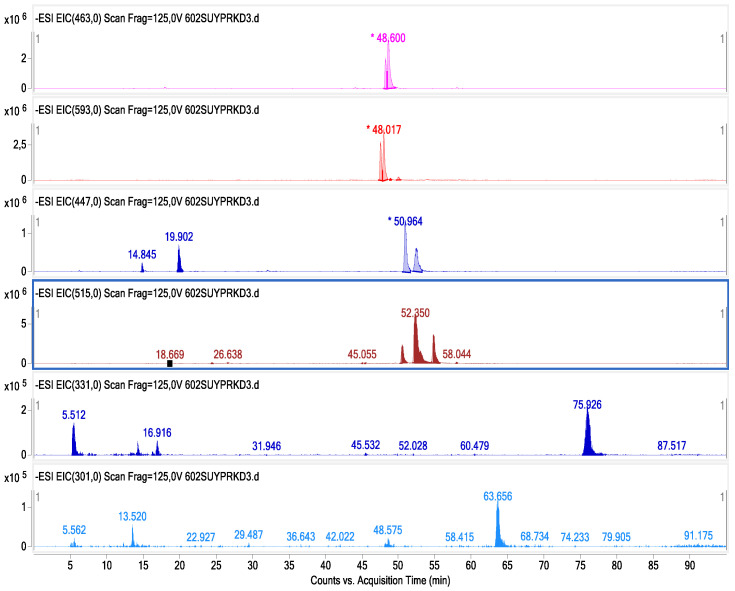
Chromatogram of phenolic compounds identified by LC-DAD and LC–ESI-MS/MS in Gw-60 °C tea infusion.

**Figure 6 foods-13-00694-f006:**
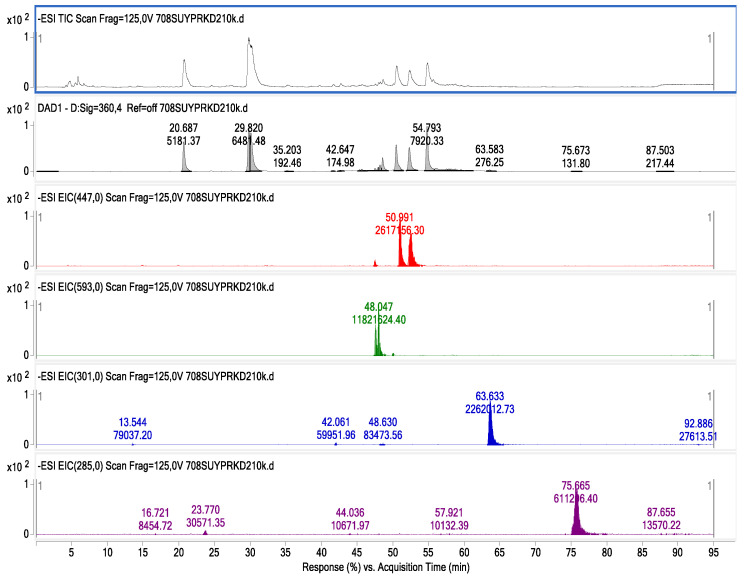
Chromatogram of phenolic compounds identified by LC-DAD and LC–ESI-MS/MS in Gw-70 °C tea infusion.

**Figure 7 foods-13-00694-f007:**
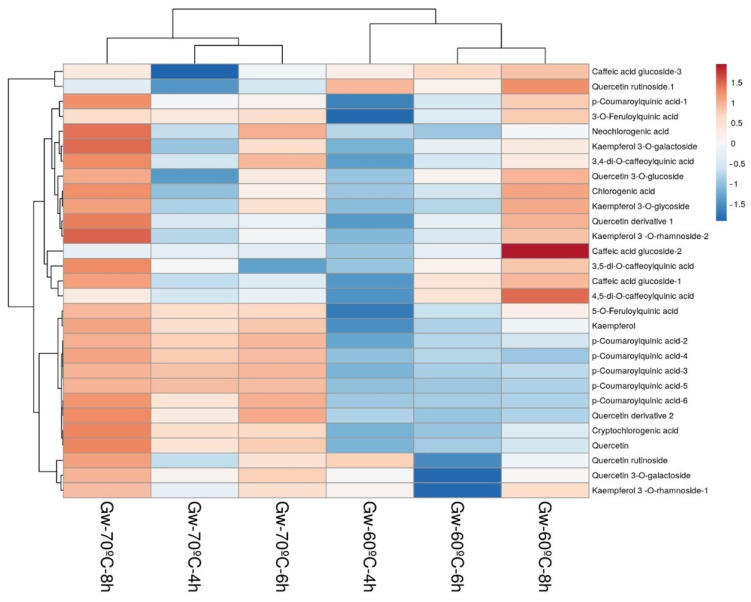
Heatmap of phenolic compounds in Guayusa water (Gw) infusions. Rows are centered; unit variance scaling is applied to rows. Both rows (29 rows; phenolics) and columns (6 columns; infusions) are clustered using correlation distance and average linkage.

**Figure 8 foods-13-00694-f008:**
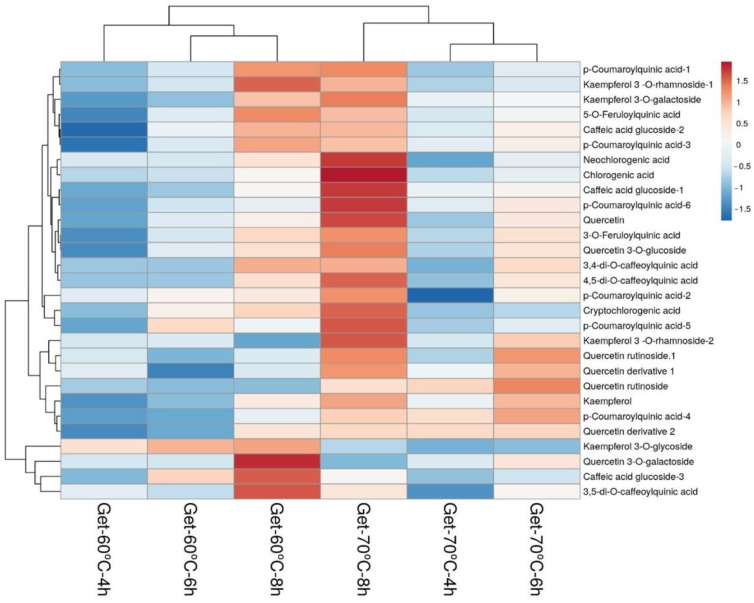
Heatmap of phenolic compounds in Guayusa ethanol-water (Get) infusions. Rows are centered; unit variance scaling is applied to rows. Both rows (29 rows; phenolics) and columns (6 columns; infusions) are clustered using correlation distance and average linkage.

**Table 1 foods-13-00694-t001:** The effect of infusion time and temperature on the colour of tea infusions.

	L*	a*	b*	C	h°
Gw-60 °C-4 h	23.43 ± 0.01 ^a^	36.17 ± 0.01 ^a^	40.28 ± 0.02 ^a^	54.13 ± 0.02 ^a^	48.08 ± 0.01 ^a^
Gw-60 °C-6 h	23.36 ± 0.01 ^a^	36.22 ± 0.01 ^a^	40.17 ± 0.05 ^a^	54.09 ± 0.04 ^a^	47.97 ± 0.04 ^b^
Gw-60 °C-8 h	23.19 ± 0.10 ^b^	36.09 ± 0.04 ^a^	39.87 ± 0.14 ^b^	53.78 ± 0.13 ^b^	47.85 ± 0.07 ^c^
Gw-70 °C-4 h	22.89 ± 0.10 ^c^	35.94 ± 0.11 ^b^	39.32 ± 0.15 ^c^	53.27 ± 0.18 ^c^	47.58 ± 0.04 ^d^
Gw-70 °C-6 h	22.52 ± 0.15 ^d^	35.80 ± 0.13 ^b^	38.69 ± 0.23 ^d^	52.71 ± 0.25 ^d^	47.22 ± 0.07 ^e^
Gw-70 °C-8 h	22.09 ± 0.02 ^e^	35.86 ± 0.02 ^b^	37.98 ± 0.06 ^e^	52.23 ± 0.05 ^e^	46.65 ± 0.04 ^f^
Get-60 °C-4 h	34.57 ± 0.01 ^a^	41.07 ± 0.01 ^c^	59.44 ± 0.04 ^a^	72.25 ± 0.03 ^a^	55.36 ± 0.02 ^a^
Get-60 °C-6 h	33.26 ± 0.04 ^b^	41.53 ± 0.04 ^b^	57.29 ± 0.09 ^b^	70.75 ± 0.09 ^b^	54.06 ± 0.03 ^b^
Get-60 °C-8 h	32.62 ± 0.03 ^c^	41.52 ± 0.04 ^b^	56.20 ± 0.13 ^c^	69.87 ± 0.13 ^c^	53.54 ± 0.04 ^c^
Get-70 °C-4 h	32.25 ± 0.04 ^d^	41.47 ± 0.03 ^b^	55.55 ± 0.06 ^d^	69.32 ± 0.07 ^d^	53.26 ± 0.01 ^e^
Get-70 °C-6 h	31.97 ± 0.15 ^e^	40.89 ± 0.13 ^d^	55.06 ± 0.29 ^e^	68.59 ± 0.31 ^e^	53.40 ± 0.06 ^d^
Get-70 °C-8 h	31.14 ± 0.01 ^f^	41.69 ± 0.02 ^a^	53.67 ± 0.01 ^f^	67.97 ± 0.02 ^f^	52.16 ± 0.02 ^f^

^a–f^ Different letters in the columns represent statistically significant differences (*p* < 0.05). Gw: Guayusa water extraction; Get: Guayusa ethanol extraction.

**Table 2 foods-13-00694-t002:** The effect of infusion time and temperature on the antioxidant capacity of infusions.

	DPPH	ABTS	TPC
	(mM TE/L)	(mM TE/L)	(mg GAE/L)
Gw-60 °C-4 h	70.40 ± 0.91 ^e^	80.65 ± 3.16 ^b^	16,152.51 ± 248.88 ^e^
Gw-60 °C-6 h	71.49 ± 1.28 ^d^	77.38 ± 2.13 ^e^	16,692.24 ± 208.04 ^d^
Gw-60 °C-8 h	73.09 ± 1.01 ^c^	78.45 ± 0.80 ^c^	18,064.84 ± 33.03 ^b^
Gw-70 °C-4 h	69.53 ± 0.51 ^f^	74.26 ± 1.29 ^f^	16,116.89 ± 63.98 ^f^
Gw-70 °C-6 h	74.41 ± 1.28 ^b^	77.73 ± 0.85 ^d^	17,310.50 ± 63.70 ^c^
Gw-70 °C-8 h	86.12 ± 1.24 ^a^	88.19 ± 1.46 ^a^	19,467.58 ± 38.49 ^a^
Get-60 °C-4 h	54.83 ± 3.00 ^f^	71.51 ± 2.86 ^f^	14,757.08 ± 37.40 ^f^
Get-60 °C-6 h	65.30 ± 1.57 ^e^	73.74 ± 2.70 ^e^	15,604.11 ± 63.93 ^e^
Get-60 °C-8 h	76.96 ± 1.17 ^c^	74.62 ± 1.65 ^d^	16,706.85 ± 42.07 ^c^
Get-70 °C-4 h	72.34 ± 2.91 ^d^	75.55 ± 0.29 ^c^	16,340.64 ± 113.30 ^d^
Get-70 °C-6 h	78.08 ± 2.09 ^b^	77.24 ± 0.06 ^b^	17,985.39 ± 63.87 ^b^
Get-70 °C-8 h	79.21 ± 1.26 ^a^	83.09 ± 2.86 ^a^	18,852.05 ± 27.89 ^a^

^a–f^ Different letters in the columns represent statistically significant differences (*p* < 0.05). Gw: Guayusa water infusions; Get: Guayusa ethanol infusions.

**Table 3 foods-13-00694-t003:** Retention time, mass spectral characteristics, and identity of phenolic compounds present in tea infusion.

Compound	RT (min)	Formula	Mass [M-H]^−^	Fragment Ions	λ max (nm)	Proposed Compound
1	17.09	C_15_H_17_O_9_	341	179, 161	324, 296sh	Caffeic acid glucoside-1
2	24.59	C_15_H_17_O_9_	341	179, 161	324, 296sh	Caffeic acid glucoside-2
3	27.9	C_15_H_17_O_9_	341	179, 161	324, 290sh	Caffeic acid glucoside-3
4	27.07	C_16_H_17_O_8_	337	191, 163	310	*p*-Coumaroylquinic acid-1
5	20.8	C_16_H_17_O_9_	353	191, 179, 135	316, 290sh	Neochlorogenic acid
6	29.67	C_16_H_17_O_9_	353	707, 191, 179	322, 296sh	Chlorogenic acid
7	35.52	C_16_H_17_O_9_	353	707, 191, 179	324, 296sh	Cryptochlorogenic acid
8	38.57	C_16_H_17_O_8_	337	191, 163	310	*p*-Coumaroylquinic acid-2
9	39.71	C_16_H_17_O_8_	337	191, 163	311	*p*-Coumaroylquinic acid-3
10	45.41	C_16_H_17_O_8_	337	191, 163	311	*p*-Coumaroylquinic acid-4
11	57.09	C_16_H_17_O_8_	337	191, 163	311	*p*-Coumaroylquinic acid-5
12	58.4	C_16_H_17_O_8_	337	191, 163	311	*p*-Coumaroylquinic acid-6
13	41.67	C_17_H_19_O_9_	367	191, 134	324, 296sh	3-*O*-Feruloylquinic acid
14	42.88	C_17_H_19_O_9_	367	191, 134	324, 296sh	5-*O*-Feruloylquinic acid
15	45.37	C_27_H_29_O_16_	609	301, 300, 271	253, 343	Quercetin rutinoside
16	45.66	C_27_H_29_O_16_	609	301, 300, 271	254, 349	Quercetin rutinoside
17	47.05	-	609	301, 300, 271	254, 349	Quercetin derivative 1
18	58.72	-	609	301, 300, 271	254, 349	Quercetin derivative 2
19	48.23	C_21_H_19_O_12_	463	301, 300, 271	254, 349	Quercetin 3-O-galactoside
20	48.57	C_21_H_19_O_12_	463	301, 300, 271	254, 351	Quercetin 3-O-glucoside
21	47.54	C_27_H_29_O_15_	593	285	264, 340	Kaempferol 3-O-rhamnoside-1
22	47.96	C_27_H_29_O_15_	593	285, 255, 227	264, 346	Kaempferol 3-O-rhamnoside-2
23	50.96	C_21_H_19_O_11_	447	342, 285, 255	264, 342	Kaempferol 3-O-galactoside
24	50.53	C_25_H_23_O_12_	515	353, 191, 179	325, 296sh	3,4-di-O-caffeoylquinic acid
25	52.49	C_21_H_19_O_11_	447	285, 255	264, 334	Kaempferol 3-O-glycoside
26	52.37	C_25_H_23_O_12_	515	353, 191, 179	326, 296sh	3,5-di-O-caffeoylquinic acid
27	54.85	C_25_H_23_O_12_	515	353, 191, 179	326, 296sh	4,5-di-O-caffeoylquinic acid
28	63.66	C_15_H_9_O_7_	301	273, 257, 179, 151	254, 349	Quercetin
29	75.93	C_15_H_9_O_6_	285	285, 229, 185, 151	264, 346	Kaempferol

**Table 4 foods-13-00694-t004:** Concentration of phenolic compounds (mg/L) in Guayusa water (Gw) infusions.

Proposed Compound	Gw-60 °C-4 h	Gw-60 °C-6 h	Gw-60 °C-8 h	Gw-70 °C-4 h	Gw-70 °C-6 h	Gw-70 °C-8 h
Caffeic acid glucoside-1	115.38 ± 0.56 ^d^	124.59 ± 0.61 ^b^	126.76 ± 1.17 ^a^	118.99 ± 0.58 ^c^	120.47 ± 0.59 ^c^	127.78 ± 0.62 ^a^
Caffeic acid glucoside-2	64.72 ± 0.32 ^c^	72.33 ± 0.35 ^b^	94.91 ± 0.88 ^a^	72.02 ± 0.15 ^b^	71.38 ± 0.18 ^b^	72.41 ± 0.30 ^b^
Caffeic acid glucoside-3	69.85 ± 0.34 ^d^	75.36 ± 0.32 ^b^	78.19 ± 0.52 ^a^	42.36 ± 0.20^f^	65.46 ± 0.22 ^e^	71.07 ± 0.21 ^c^
*p*-Coumaroylquinic acid-1	46.98 ± 0.23 ^f^	60.25 ± 0.29 ^e^	75.50 ± 0.70 ^b^	65.53 ± 0.12 ^d^	67.67 ± 0.18 ^c^	80.60 ± 0.19 ^a^
Neochlorogenic acid	3462.19 ± 10.90 ^b^	3452.12 ± 13.85 ^b^	3503.15 ± 12.31 ^b^	3466.74 ± 12.92 ^b^	3563.44 ± 10.40 ^a^	3587.26 ± 14.21 ^a^
Chlorogenic acid	6428.28 ± 31.38 ^b^	6447.19 ± 31.47 ^ab^	6547.63 ± 60.38 ^a^	6422.63 ± 31.35 ^ab^	6494.11 ± 31.70 ^ab^	6557.41 ± 32.01 ^a^
Cryptochlorogenic acid	225.42 ± 1.10 ^f^	238.84 ± 1.17 ^e^	280.10 ± 2.58 ^d^	342.01 ± 1.67 ^c^	348.82 ± 1.70 ^b^	395.38 ± 1.93 ^a^
*p*-Coumaroylquinic acid-2	40.77 ± 0.20 ^f^	51.23 ± 0.25 ^e^	55.33 ± 0.51 ^c^	83.55 ± 0.41 ^c^	86.89 ± 0.42 ^b^	88.49 ± 0.43 ^a^
*p*-Coumaroylquinic acid-3	56.86 ± 0.28 ^e^	62.63 ± 0.31 ^d^	65.14 ± 0.60 ^c^	96.99 ± 0.47 ^b^	98.64 ± 0.48 ^a b^	99.39 ± 0.49 ^a^
*p*-Coumaroylquinic acid-4	26.61 ± 0.13 ^f^	34.27 ± 0.17 ^d^	29.31 ± 0.27 ^e^	83.46 ± 0.41 ^c^	86.80 ± 0.42 ^b^	92.83 ± 0.45 ^a^
*p*-Coumaroylquinic acid-5	12.24 ± 0.06 ^e^	17.28 ± 0.08 ^d^	25.24 ± 0.23 ^c^	148.75 ± 0.73 ^b^	148.08 ± 0.42 ^b^	154.93 ± 0.76 ^a^
*p*-Coumaroylquinic acid-6	17.22 ± 0.08 ^f^	20.66 ± 0.10 ^e^	29.86 ± 0.28 ^d^	140.98 ± 0.69 ^c^	186.79 ± 0.91 ^b^	204.35 ± 1.00 ^a^
3-*O*-Feruloylquinic acid	70.69 ± 0.35 ^e^	100.38 ± 0.49 ^d^	123.33 ± 1.14 ^a^	114.56 ± 0.56 ^c^	118.28 ± 0.58 ^b^	119.38 ± 0.58 ^b^
5-*O*-Feruloylquinic acid	72.51 ± 0.23 ^e^	94.77 ± 0.46 ^d^	114.60 ± 1.06 ^c^	121.61 ± 0.59 ^b^	123.54 ± 0.60 ^b^	130.08 ± 0.64 ^a^
Quercetin rutinoside	40.20 ± 0.20 ^b^	36.27 ± 0.18 ^e^	38.74 ± 0.36 ^c^	37.71 ± 0.18 ^b^	39.77 ± 0.19 ^b^	40.82 ± 0.20 ^a^
Quercetin rutinoside	61.92 ± 0.30 ^b^	58.99 ± 0.29 ^c^	63.04 ± 0.58 ^a^	53.15 ± 0.26 ^e^	56.49 ± 0.28 ^d^	57.15 ± 0.28 ^d^
Quercetin derivative 1	15.14 ± 0.07 ^e^	17.51 ± 0.09 ^c^	20.04 ± 0.18 ^b^	17.06 ± 0.08 ^d^	17.62 ± 0.09 ^c^	20.83 ± 0.10 ^a^
Quercetin derivative 2	38.28 ± 0.19 ^d^	35.98 ± 0.18 ^e^	38.37 ± 0.35 ^d^	54.08 ± 0.26 ^c^	65.62 ± 0.32 ^b^	68.77 ± 0.34 ^a^
Quercetin 3-O-galactoside	161.02 ± 0.79 ^b^	146.93 ± 0.72 ^c^	161.30 ± 1.49 ^b^	161.95 ± 0.79 ^b^	166.65 ± 0.81 ^a^	168.48 ± 0.82 ^a^
Quercetin 3-O-glucoside	401.32 ± 1.96 ^c^	422.93 ± 2.06 ^b^	440.25 ± 4.06 ^a^	392.52 ± 1.92 ^c^	426.33 ± 2.08 ^b^	441.89 ± 2.16 ^a^
Kaempferol 3-O-rhamnoside-1	65.85 ± 0.32 ^d^	54.57 ± 0.27 ^e^	68.61 ± 0.63 ^b^	63.99 ± 0.31 ^d^	68.33 ± 0.33 ^b^	70.33 ± 0.34 ^a^
Kaempferol 3-O-rhamnoside-2	73.46 ± 0.36 ^e^	76.78 ± 0.37 ^d^	84.12 ± 0.78 ^b^	75.12 ± 0.37 ^d^	79.07 ± 0.39 ^c^	87.77 ± 0.43 ^a^
Kaempferol 3-O-galactoside	172.75 ± 0.84 ^f^	212.74 ± 1.04 ^d^	236.22 ± 2.18 ^c^	181.57 ± 0.89 ^e^	248.58 ± 1.21 ^b^	287.40 ± 1.40 ^a^
3,4-di-O-caffeoylquinic acid	2540.74 ± 12.40 ^d^	2568.04 ± 12.54 ^cd^	2596.23 ± 23.94 ^abc^	2567.76 ± 12.54 ^cd^	2618.47 ± 12.78 ^ab^	2631.11 ± 12.84 ^a^
Kaempferol 3-O-glycoside	95.00 ± 0.46 ^e^	100.63 ± 0.49 ^d^	140.20 ± 1.29 ^b^	100.08 ± 0.49 ^d^	128.28 ± 0.63 ^c^	142.37 ± 0.70 ^a^
3,5-di-O-caffeoylquinic acid	2298.18 ± 7.22 ^c^	2351.86 ± 10.48 ^b^	2382.19 ± 13.97 ^ab^	2344.89 ± 10.11 ^b^	2283.03 ± 11.02 ^c^	2403.95 ± 8.74 ^a^
4,5-di-O-caffeoylquinic acid	3450.31 ± 16.84 ^d^	3626.28 ± 17.70 ^bc^	3718.92 ± 34.30 ^a^	3533.06 ± 17.25 ^c^	3565.06 ± 17.40 ^c^	3609.36 ± 17.62 ^bc^
Quercetin	60.83 ± 0.30 ^f^	63.31 ± 0.31 ^e^	67.01 ± 0.62 ^d^	79.05 ± 0.39 ^c^	82.75 ± 0.40 ^b^	88.88 ± 0.43 ^a^
Kaempferol	41.40 ± 0.20 ^f^	47.82 ± 0.23 ^e^	54.21 ± 0.50 ^d^	60.60 ± 0.30 ^c^	63.03 ± 0.31 ^b^	65.43 ± 0.32 ^a^
Total	20.226.12 ± 98.74 ^d^	20.672.56 ± 100.92 ^c^	21.258.52 ± 196.05 ^b^	21.042.79 ± 102.73 ^b^	21.489.44 ± 104.91 ^b^	21.965.91 ± 107.24 ^a^

^a–f^ Different letters in the rows represent statistically significant differences (*p* < 0.05). Gw: Guayusa water infusions.

**Table 5 foods-13-00694-t005:** The concentration of phenolic compounds (mg/L) in Guayusa ethanol-water (Get) infusions.

Proposed Compound	Get-60 °C-4 h	Get-60 °C-6 h	Get-60 °C-8 h	Get-70 °C-4 h	Get-70 °C-6 h	Get-70 °C-8 h
Caffeic acid glucoside-1	117.10 ± 0.57 ^d^	119.47 ± 0.58 ^d^	127.28 ± 0.62 ^b^	125.17 ± 0.61 ^c^	127.58 ± 0.62 ^b^	140.14 ± 1.24 ^a^
Caffeic acid glucoside-2	47.39 ± 0.23 ^e^	77.19 ± 0.38 ^c^	98.22 ± 0.24 ^a^	73.04 ± 0.36 ^d^	84.02 ± 0.41 ^b^	97.59 ± 0.87 ^a^
Caffeic acid glucoside-3	63.25 ± 0.31 ^e^	86.99 ± 0.42 ^b^	98.15 ± 0.48 ^a^	64.30 ± 0.31 ^e^	69.83 ± 0.34 ^d^	79.12 ± 0.70 ^c^
*p*-Coumaroylquinic acid-1	73.28 ± 0.36 ^d^	81.42 ± 0.40 ^c^	108.17 ± 0.53 ^a^	74.93 ± 0.37 ^d^	84.20 ± 0.41 ^b^	109.50 ± 0.97 ^a^
Neochlorogenic acid	3067.05 ± 11.97 ^c^	3060.35 ± 14.94 ^c^	3176.38 ± 12.51 ^b^	2979.04 ± 14.54 ^d^	3091.19 ± 12.09 ^c^	3319.35 ± 23.45 ^a^
Chlorogenic acid	4484.53 ± 21.89 ^d^	4518.18 ± 22.06 ^d^	4688.97 ± 22.89 ^b^	4496.08 ± 21.95 ^d^	4606.45 ± 22.49 ^c^	5145.29 ± 25.66 ^a^
Cryptochlorogenic acid	239.05 ± 1.17 ^d^	261.03 ± 1.27 ^c^	270.03 ± 1.32 ^b^	240.83 ± 1.18 ^d^	244.00 ± 1.19 ^d^	284.67 ± 2.53 ^a^
*p*-Coumaroylquinic acid-2	105.82 ± 0.52 ^d^	110.94 ± 0.24 ^c^	113.39 ± 0.55 ^b^	87.01 ± 0.32 ^e^	111.89 ± 0.55 ^b c^	124.50 ± 1.10 ^a^
*p*-Coumaroylquinic acid-3	104.69 ± 0.51 ^d^	111.01 ± 0.54 ^c^	118.43 ± 0.58 ^a^	111.83 ± 0.35 ^c^	114.38 ± 0.56 ^b^	117.24 ± 1.04 ^a^
*p*-Coumaroylquinic acid-4	52.82 ± 0.26 ^f^	56.50 ± 0.28 ^e^	83.48 ± 0.41 ^d^	106.27 ± 0.51 ^c^	120.35 ± 0.59 ^a^	110.14 ± 0.98 ^b^
*p*-Coumaroylquinic acid-5	85.42 ± 0.42 ^f^	114.05 ± 0.56 ^b^	103.37 ± 0.50 ^c^	91.63 ± 0.25 ^e^	99.34 ± 0.48 ^d^	129.30 ± 1.15 ^a^
*p*-Coumaroylquinic acid-6	79.73 ± 0.39 ^f^	94.42 ± 0.46 ^e^	102.57 ± 0.28 ^c^	99.97 ± 0.49 ^d^	114.81 ± 0.56 ^b^	144.74 ± 1.28 ^a^
3-*O*-Feruloylquinic acid	85.88 ± 0.42 ^f^	101.29 ± 0.49 ^d^	119.94 ± 0.59 ^b^	97.55 ± 0.48 ^e^	117.50 ± 0.57 ^c^	129.19 ± 1.15 ^a^
5-*O*-Feruloylquinic acid	103.91 ± 0.51 ^f^	133.89 ± 0.65 ^d^	174.87 ± 0.85 ^a^	129.25 ± 0.63 ^e^	141.03 ± 0.69 ^c^	165.17 ± 1.47 ^b^
Quercetin rutinoside	29.61 ± 0.14 ^c^	28.74 ± 0.14 ^d^	28.67 ± 0.14 ^d^	38.07 ± 0.19 ^b^	41.46 ± 0.20 ^a^	37.31 ± 0.33 ^b^
Quercetin rutinoside	65.67 ± 0.32 ^b^	60.73 ± 0.30 ^d^	66.54 ± 0.32 ^b^	63.12 ± 0.31 ^c^	79.58 ± 0.39 ^a^	80.36 ± 0.71 ^a^
Quercetin derivative 1	24.12 ± 0.12 ^d^	19.36 ± 0.09 ^e^	23.74 ± 0.12 ^d^	25.05 ± 0.12 ^c^	29.12 ± 0.14 ^b^	29.94 ± 0.27 ^a^
Quercetin derivative 2	60.26 ± 0.29 ^d^	63.96 ± 0.31 ^c^	83.22 ± 0.41 ^b^	85.41 ± 0.42 ^a^	86.44 ± 0.42 ^a^	85.99 ± 0.76 ^a^
Quercetin 3-O-galactoside	164.21 ± 0.80 ^c^	163.69 ± 0.80 ^c^	184.80 ± 0.90 ^a^	165.21 ± 0.81 ^c^	172.85 ± 0.84 ^b^	159.35 ± 1.41 ^d^
Quercetin 3-O-glucoside	379.63 ± 1.85 ^e^	420.77 ± 2.05 ^c^	451.28 ± 2.20 ^b^	405.73 ± 1.98 ^d^	448.43 ± 2.19 ^b^	479.54 ± 4.26 ^a^
Kaempferol 3-O-rhamnoside-1	71.39 ± 0.35 ^e^	76.20 ± 0.37 ^c^	97.61 ± 0.48 ^a^	73.66 ± 0.36 ^d^	77.29 ± 0.38 ^c^	92.13 ± 0.82 ^b^
Kaempferol 3-O-rhamnoside-2	85.75 ± 0.42 ^d^	87.13 ± 0.43 ^c^	74.58 ± 0.36 ^e^	85.51 ± 0.42 ^d^	105.91 ± 0.52 ^b^	117.92 ± 1.05 ^a^
Kaempferol 3-O-galactoside	250.18 ± 1.22 ^e^	260.85 ± 1.27 ^d^	313.65 ± 1.53 ^b^	284.10 ± 1.39 ^c^	288.41 ± 1.41 ^c^	328.08 ± 2.91 ^a^
3,4-di-O-caffeoylquinic acid	2463.61 ± 12.03 ^c^	2465.04 ± 12.03 ^c^	2663.39 ± 13.00 ^a^	2439.60 ± 11.91 ^c^	2622.91 ± 12.80 ^b^	2665.55 ± 23.65 ^a^
Kaempferol 3-O-glycoside	190.16 ± 0.93 ^c^	212.53 ± 1.04 ^b^	217.93 ± 1.06 ^a^	111.33 ± 0.54 ^f^	115.75 ± 0.57 ^e^	128.72 ± 1.14 ^d^
3,5-di-O-caffeoylquinic acid	2162.81 ± 10.56 ^c^	2138.01 ± 10.44 ^c^	2286.31 ± 11.16 ^a^	2085.66 ± 10.18 ^d^	2188.69 ± 10.68 ^b^	2209.15 ± 19.60 ^b^
4,5-di-O-caffeoylquinic acid	3524.99 ± 17.21 ^c^	3531.31 ± 17.24 ^c^	3803.41 ± 18.57 ^b^	3516.01 ± 17.16 ^c^	3772.46 ± 18.42 ^b^	3977.99 ± 35.30 ^a^
Quercetin	70.65 ± 0.34 ^f^	80.02 ± 0.39 ^d^	85.83 ± 0.42 ^c^	74.14 ± 0.36 ^e^	87.44 ± 0.43 ^b^	101.24 ± 0.90 ^a^
Kaempferol	41.09 ± 0.20 ^f^	44.03 ± 0.21 ^e^	52.90 ± 0.26 ^c^	49.71 ± 0.24 ^d^	57.10 ± 0.28 ^b^	58.15 ± 0.52 ^a^
Total	18.294.08 ± 89.31 ^d^	18.579.09 ± 90.70 ^d^	19.817.10 ± 96.75 ^b^	18.279.20 ± 89.24 ^d^	19.300.41 ± 94.22 ^c^	20.647.34 ± 183.21 ^a^

^a–f^ Different letters in the rows represent statistically significant differences (*p* < 0.05). Get: Guayusa ethanol infusions.

**Table 6 foods-13-00694-t006:** Antimicrobial effects of Guayusa infusions.

	Inhibition Zone Diameter (mm)	
Test Microorganisms	Gw-70 °C-8 h	Get-70 °C-8 h
*S. aureus* ATCC 29213	21.27 ± 0.42 ^a^	17.40 ± 0.26 ^b^
*B. subtilis* ATCC 11774	17.75 ± 0.46 ^a^	14.51 ± 0.40 ^b^
*K. pneumoniae* ATCC 13883	20.98 ± 0.51 ^a^	14.19 ± 0.62 ^b^
*E. coli* ATCC 25922	-	-

^a,b^ Different letters in the rows represent statistically significant differences (*p* < 0.05).

**Table 7 foods-13-00694-t007:** Effect of in vitro digestion model on antioxidant activity (DPPH and ABTS) and total phenolic compounds (TPC).

	ABTS (mM/L)	DPPH (mM/L)	TPC (mg/L)
Gw-70 °C-8 h	34.14 ± 1.96 ^d^	36.63 ± 1.74 ^d^	9468.42 ± 103.53 ^cd^
Oral	37.82 ± 1.84 ^cd^	51.58 ± 0.08 ^c^	9293.86 ± 115.91 ^d^
Gastric	45.72 ± 1.11 ^b^	61.84 ± 0.88 ^b^	12.352.63 ± 131.89 ^b^
Intestinal	58.42 ± 1.90 ^a^	66.81 ± 1.26 ^a^	14.856.14 ± 59.85 ^a^
Get-70 °C-8 h	37.69 ± 2.26 ^e^	45.08 ± 0.83 ^e^	10.340.35 ± 83.73 ^e^
Oral	43.34 ± 0.86 ^cd^	61.04 ± 1.02 ^c^	11.882.46 ± 98.62 ^d^
Gastric	46.64 ± 2.85 ^bc^	67.08 ± 1.27 ^b^	14.114.04 ± 368.44 ^cd^
Intestinal	61.36 ± 2.86 ^a^	70.01 ± 1.94 ^a^	17.112.28 ± 220.88 ^a^

^a–e^ Different letters in the columns represent statistically significant differences (*p* < 0.05).

**Table 8 foods-13-00694-t008:** Effect of in vitro digestion model on phenolic compounds (mg/L).

		Ethanol Infusions						Water Infusions						
Proposed Compound	Initial	Oral	%	Gastric	%	Intestinal	%	Initial	Oral	%	Gastric	%	Intestinal	%
Caffeic acid glucoside-1	140.14 ± 1.24 ^c^	107.58 ± 0.59 ^d^	−23.2	120.10 ± 1.00 ^e^	−14.3	144.08 ± 2.78 ^a^	2.8	127.78 ± 0.62 ^d^	119.06 ± 1.33 ^e^	−6.8	127.25 ± 4.17 ^f^	−0.4	147.97 ± 1.18 ^a^	15.8
Caffeic acid glucoside-2	97.59 ± 0.87 ^e^	85.60 ± 0.56 ^f^	−12.3	104.87 ± 2.16 ^d^	7.5	124.70 ± 0.80 ^a^	27.8	72.41 ± 0.30 ^g^	65.06 ± 0.77 ^h^	−10.2	109.72 ± 2.71 ^c^	51.5	122.04 ± 1.64 ^b^	68.5
Caffeic acid glucoside-3	79.12 ± 0.70 ^e^	77.25 ± 1.07 ^f^	−2.4	86.51 ± 1.12 ^d^	9.3	112.57 ± 1.45 ^c^	42.3	71.07 ± 0.21 ^h^	73.68 ± 1.01 ^g^	3.7	116.08 ± 2.15 ^b^	63.3	135.46 ± 0.48 ^a^	90.6
p-Coumaroylquinic acid-1	109.50 ± 0.97 ^d^	62.07 ± 1.31 ^h^	−43.3	106.48 ± 0.88 ^e^	−2.8	122.45 ± 2.20 ^c^	11.8	80.60 ± 0.19 ^f^	69.16 ± 1.19 ^g^	−14.2	129.78 ± 2.57 ^b^	61.0	152.33 ± 0.39 ^a^	89.0
Neochlorogenic acid	3319.35 ± 23.45 ^f^	3137.15 ± 2.61 ^h^	−5.5	3556.73 ± 7.88 ^e^	7.2	4143.54 ± 3.42 ^b^	24.8	3587.26 ± 14.21 ^d^	3315.00 ± 1.42 ^g^	−7.6	3592.75 ± 3.18 ^c^	0.2	4327.97 ± 1.46 ^a^	20.6
Chlorogenic acid	5145.29 ± 25.66 ^g^	4909.03 ± 2.78 ^h^	−4.6	5270.98 ± 2.49 ^f^	2.4	6791.18 ± 2.17 ^c^	32.0	6557.41 ± 32.01 ^d^	6413.88 ± 72.29 ^e^	−2.2	6822.25 ± 2.11 ^b^	4.0	7882.65 ± 8.02 ^a^	20.2
Cryptochlorogenic acid	284.67 ± 2.53 ^f^	205.23 ± 4.71 ^h^	−27.9	247.99 ± 1.15 ^g^	−12.9	327.07 ± 2.03 ^e^	14.9	395.38 ± 1.93 ^b^	388.08 ± 2.72 ^d^	−1.8	394.93 ± 1.51 ^a^	−0.1	424.28 ± 0.96 ^a^	7.3
p-Coumaroylquinic acid-2	124.50 ± 1.10 ^c^	102.99 ± 0.65 ^d^	−17.3	136.82 ± 1.10 ^b^	9.9	165.76 ± 3.03 ^a^	33.1	88.49 ± 0.43 ^g^	88.87 ± 1.60 ^g^	0.4	94.22 ± 1.16 ^f^	6.5	115.88 ± 1.32 ^d^	31.0
p-Coumaroylquinic acid-3	117.24 ± 1.04 ^c^	100.34 ± 2.35 ^e^	−14.4	119.79 ± 2.80 ^b^	2.2	186.48 ± 2.29 ^a^	59.1	99.39 ± 0.49 ^f^	89.82 ± 1.67 ^g^	−9.6	106.21 ± 1.01 ^d^	6.9	119.95 ± 1.18 ^b^	20.7
p-Coumaroylquinic acid-4	110.14 ± 0.98 ^c^	97.12 ± 1.25 ^f^	−11.8	115.93 ± 1.25 ^b^	5.3	142.98 ± 3.60 ^a^	29.8	92.83 ± 0.45 ^g^	91.55 ± 0.93 ^h^	−1.4	100.12 ± 0.26 ^e^	7.9	108.29 ± 0.72 ^d^	16.7
p-Coumaroylquinic acid-5	129.30 ± 1.15 ^f^	82.07 ± 1.31 ^h^	−36.5	115.86 ± 0.20 ^g^	−10.4	134.28 ± 1.08 ^e^	3.9	154.93 ± 0.76 ^d^	168.57 ± 2.02 ^c^	8.8	181.13 ± 4.26 ^b^	16.9	185.09 ± 0.06 ^a^	19.5
p-Coumaroylquinic acid-6	144.74 ± 1.28 ^f^	89.29 ± 1.00 ^h^	−38.3	136.80 ± 3.57 ^g^	−5.5	165.60 ± 1.75 ^e^	14.4	204.35 ± 1.00 ^c^	197.06 ± 1.33 ^d^	−3.6	244.12 ± 1.98 ^b^	19.5	257.78 ± 0.87 ^a^	26.1
3-O-Feruloylquinic acid	129.19 ± 1.15 ^c^	127.03 ± 1.37 ^c d^	−1.7	145.67 ± 1.03 ^b^	12.8	163.48 ± 1.41 ^a^	26.5	119.38 ± 0.58 ^e^	118.32 ± 0.96 ^d^	−0.9	121.56 ± 1.19 ^d^	1.8	147.34 ± 0.94 ^b^	23.4
5-O-Feruloylquinic acid	165.17 ± 1.47 ^b^	126.59 ± 1.99 ^g^	−23.4	154.27 ± 2.87 ^c^	−6.6	184.70 ± 1.38 ^a^	11.8	130.08 ± 0.64 ^f^	126.24 ± 1.07 ^g^	−3.0	133.02 ± 0.90 ^f^	2.3	151.02 ± 0.25 ^d^	16.1
Quercetin rutinoside	37.31 ± 0.33 ^c^	34.21 ± 0.41 ^d^	−8.3	32.78 ± 1.16 ^e^	−1.1	25.43 ± 0.79 ^e^	−31.8	40.82 ± 0.20 ^a^	39.49 ± 0.72 ^b^	−3.3	36.02 ± 0.02 ^c^	−11.8	28.71 ± 0.51 ^f^	−29.7
Quercetin rutinoside	80.36 ± 0.71 ^a^	79.92 ± 0.12 ^a^	−0.5	75.96 ± 0.77 ^b^	−5.5	74.18 ± 1.66 ^b^	−7.7	57.15 ± 0.28 ^c^	55.27 ± 0.33 ^d^	−3.3	51.30 ± 0.65 ^e^	−10.2	45.74 ± 1.49 ^f^	−20.0
Quercetin derivative 1	29.94 ± 0.27 ^a^	28.80 ± 0.28 ^a^	−3.8	25.07 ± 0.75 ^b^	−16.3	22.70 ± 0.88 ^c^	−24.2	20.83 ± 0.10 ^d^	19.26 ± 0.37 ^e^	−7.5	18.67 ± 0.75 ^f^	−10.4	18.83 ± 0.30 ^f^	−9.6
Quercetin derivative 2	85.99 ± 0.76 ^a^	85.24 ± 1.08 ^a^	−0.9	82.02 ± 1.10 ^b^	−4.6	80.95 ± 0.42 ^c^	−5.9	68.77 ± 0.34 ^d^	66.28 ± 0.40 ^e^	−3.6	62.85 ± 0.21 ^f^	−8.6	58.73 ± 0.61 ^g^	−14.6
Quercetin 3-O-galactoside	159.35 ± 1.41 ^b^	134.48 ± 0.74 ^d^	−15.6	123.54 ± 0.65 ^e^	−22.5	125.15 ± 1.61 ^e^	−21.5	168.48 ± 0.82 ^a^	159.26 ± 1.05 ^b^	−5.5	146.74 ± 0.66 ^c^	−12.9	133.36 ± 0.58 ^d^	−20.8
Quercetin 3-O-glucoside	479.54 ± 4.26 ^a^	464.11 ± 1.26 ^b^	−3.2	374.90 ± 1.55 ^f^	−21.8	366.96 ± 1.21 ^g^	−23.5	441.89 ± 2.16 ^c^	437.51 ± 0.86 ^d^	−1.0	393.60 ± 0.70 ^e^	−10.9	366.84 ± 0.64 ^g^	−17.0
Kaempferol 3-O-rhamnoside-1	92.13 ± 0.82 ^a^	87.80 ± 0.28 ^b^	−4.7	65.79 ± 0.87 ^d^	−28.6	60.72 ± 1.07 ^e^	−34.1	70.33 ± 0.34 ^c^	64.62 ± 0.54 ^d^	−8.1	59.53 ± 0.66 ^e^	−15.4	55.66 ± 0.71 ^f^	−20.9
Kaempferol 3-O-rhamnoside-2	117.92 ± 1.05 ^a^	106.36 ± 0.91 ^b^	−9.8	90.16 ± 1.47 ^d^	−23.5	95.20 ± 1.03 ^c^	−19.3	87.77 ± 0.43 ^e^	82.32 ± 0.45 ^f^	−6.2	77.75 ± 0.92 ^g^	−11.4	63.30 ± 0.08 ^h^	−27.9
Kaempferol 3-O-galactoside	328.08 ± 2.91 ^a^	273.94 ± 0.09 ^d^	−16.5	237.67 ± 1.03 ^g^	−27.6	226.90 ± 1.42 ^h^	−30.8	287.40 ± 1.40 ^b^	280.20 ± 1.12 ^c^	−2.5	262.94 ± 1.50 ^e^	−8.5	253.73 ± 0.52 ^f^	−11.7
3,4-di-O-caffeoylquinic acid	2665.55 ± 23.65 ^a^	2209.55 ± 3.47 ^d^	−17.1	2141.66 ± 1.00 ^e^	−19.7	2048.97 ± 2.72 ^g^	−23.1	2631.11 ± 12.84 ^a^	2478.75 ± 1.76 ^b^	−5.8	2390.81 ± 1.68 ^c^	−9.1	2093.59 ± 0.58 ^f^	−20.4
Kaempferol 3-O-glycoside	128.72 ± 1.14 ^c^	118.29 ± 1.00 ^d^	−8.1	108.97 ± 1.18 ^f^	−15.3	100.97 ± 0.71 ^g^	−21.6	142.37 ± 0.70 ^a^	132.04 ± 0.59 ^b^	−7.3	128.40 ± 1.69 ^c^	−9.8	110.87 ± 0.67 ^e^	−22.1
3,5-di-O-caffeoylquinic acid	2209.15 ± 19.60 ^b^	2067.49 ± 2.14 ^d^	−6.4	1793.84 ± 0.62 ^f^	−18.8	1613.11 ± 2.10 ^g^	−27.0	2403.95 ± 8.74 ^a^	2152.15 ± 1.20 ^c^	−10.5	1901.43 ± 0.66 ^e^	−20.9	1740.44 ± 0.48 ^f^	−27.6
4,5-di-O-caffeoylquinic acid	3977.99 ± 35.30 ^a^	3770.45 ± 2.19 ^b^	−5.2	3645.80 ± 1.99 ^c^	−8.4	3238.45 ± 3.72 ^g^	−18.6	3609.36 ± 17.62 ^d^	3486.66 ± 15.08 ^e^	−3.4	3439.14 ± 1.85 ^f^	−4.7	3221.54 ± 0.47 ^h^	−10.7
Total	20.487.9 ± 22.9 ^f^	18.769.97 ± 37.5 ^h^	−8.4	19.216.95 ± 37.8 ^g^	−6.2	20.988.60 ± 47.1 ^d^	2.4	21.811.60 ± 22.45 ^b^	20.778.16 ± 81.55 ^e^	−4.7	21.242.31 ± 35.2 ^c^	−2.6	22.469.37 ± 26.9 ^a^	3.0

^a–h^ Different letters in the columns represent statistically significant differences (*p* < 0.05). % values show the change in digestion stages according to the initial concentration.

## Data Availability

The original contributions presented in the study are included in the article, further inquiries can be directed to the corresponding author.

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
