# Peer review of "Exploring the Impact of Infusion Parameters and In Vitro Digestion on the Phenolic Profile and Antioxidant Capacity of Guayusa (Ilex guayusa Loes.) Tea Using Liquid Chromatography, Diode Array Detection, and Electrospray Ionization Tandem Mass Spectrometry"

_foods, 2024, doi:10.3390/foods13050694_

Round 1

Reviewer 1 Report

Comments and Suggestions for Authors

The article presents the results regarding the content of bioactive compounds and biological properties of Guayusa Tea. The presented properties were described during and after simulated in vitro digestion and depending on brewing conditions.

Article matches Aims and Scope of the journal.

The article submitted for review requires corrections, a minor revision.

Below are detailed comments and questions:

1. Chapter 3.2: Please add a discussion comparing the obtained antioxidant capacity results with Camellia sinensis tea. Do Guayusa Tea infusions have a higher, comparable or lower content of polyphenol compounds and antioxidant capacity?

2. Table 2 and 5:  Please complete the missing homogeneous groups and standardize the format.

3. Please standardize the m/z format in italics throughout the text

4. Please add - in the missing places - next to [M-H]

5. Table 3,4, 5 and 8: Please remove the space next to kaempferol

6. Throughout the text: please change the names of microorganisms and 'i vitro' to italics

Author Response

Dear Reviewer,

We would like to express our sincere gratitude for taking the time to review our manuscript. Your insightful comments and constructive feedback have been invaluable to us. We have carefully considered each point you raised and provided detailed responses in our revision. Your expertise and thorough review have undoubtedly enhanced the quality of our work. Thank you once again for your time and assistance; we truly appreciate it.

Please see our responses attached. 

Reviewer 2 Report

Comments and Suggestions for Authors

This submission reports the results of a study based on two different solvents used to evaluate the extraction of antioxidants and their implications for in vitro study.

Although the experimental design is correct and the results are critically discussed, I cannot find elements of innovation except in the tea species studied.

Furthermore, there is no clear explanation for the choice of ethanol extraction compared to water.

In my opinion this can be considered more like a case-report.

Comments on the Quality of English Language

English is fine, only very minor editing of English language are required

Author Response

(The authors gave the same response as above.)

Reviewer 3 Report

Comments and Suggestions for Authors

1 for brewing tea, water should be used as the solvent. Why is ethanol also applied in this study? Are they drinkable that tea extracts from ethanol?

2 why are those infusion parameters chosen in sample preparations? The long infusion time is maybe unsuitable for both consumer and industry process.  Why not applied higher infusion temperature?

3 so many phenolic isomers were reported and listed in the Tables. Are you sure about they are isomers for their totally same MS/MS fragments and UV spectra, such as peak 13 and 14, peak 15 and 16, peak 21 and 22.

4 what did you want to express with Heatmap analysis in Figure 7 and 8?

Author Response

(The authors gave the same response as above.)

Reviewer 4 Report

Comments and Suggestions for Authors

Comments and Suggestions for Authors

The overall study design is clear, and the aims have been achieved, however, several major issues need to be addressed.

Abstract.

- Background?

- Conclusions: indicate the main conclusions or interpretations.

Keywords.

- Include words other than those included in the title.

Introduction.

- In the introduction section the authors could add more background information regarding how different stages of digestion may influence the antioxidant capacity and total phenolic content of Guayusa (Ilex guayusa Loes.) Tea. In addition, the authors can introduce ABTS and DPPH with more details and add their full terms.

- The word GRAS is the first time it is mentioned, you need to explain the abbreviation “GRAS”.

- The authors did not pose a hypothesis. Add hypothesis.

Materials and Methods.

- Lines 79, 128, 135, 186, 376: hours should be h.

- Line 117: μl should be μL.

- Lines 117, 119: ml should be mL.

- Lines 118, 120: minutes should be min.

2.1. Standards and chemicals. Enzymes?

2.5. Antioxidant capacity (ABTS, DPPH): Expression of results?, standard?

- Line 377, 407: in vitro should be italics.

- Lines 392, 394: E. coli should be italics.

- Table 6 should include statistical analysis by column (Gw-70°-8h Get-70°-8h) and discuss in those terms.

- Line 414, 418: Trolox/Lin should be Trolox/L in.

- Table 8. “Total” should include statistical analysis by column (initial, oral, gastric, intestinal).

- 3.5. Impact of in vitro digestion on bioactive compound profiles. The authors attribute the changes in antioxidant capacity and total phenolic content only to changes in pH, what about enzymes? How do they influence? Explain more regarding how these factors may influence the digestive process.

More discussion and analysis are required.

Conclusions

- What future work is recommended?

- Recommendations to improve the process?

- What application does this study have?

Author Response

(The authors gave the same response as above.)

Round 2

Reviewer 2 Report

Comments and Suggestions for Authors

Authors responded to the comments and integrated the text.

The sumission can be accepted

Reviewer 4 Report

Comments and Suggestions for Authors Authors have made my suggestions